# HP1-Mediated Silencing of the *Doublesex1* Gene for Female Determination in the Crustacean *Daphnia magna*

**DOI:** 10.3390/jdb13030023

**Published:** 2025-07-03

**Authors:** Junya Leim, Nikko Adhitama, Quang Dang Nong, Pijar Religia, Yasuhiko Kato, Hajime Watanabe

**Affiliations:** 1Department of Biotechnology, Graduate School of Engineering, The University of Osaka, 2-1 Yamadaoka, Suita 565-0871, Osaka, Japan; 2Institute for Open and Transdisciplinary Research Initiatives (OTRI), The University of Osaka, Suita 565-0871, Osaka, Japan

**Keywords:** environmental sex determination, HP1, gene silencing, *doublesex1*, *Daphnia magna*

## Abstract

The crustacean *Daphnia magna* produces genetically identical females and males by parthenogenesis. Males are produced in response to environmental cues including crowding and lack of food. For male development, the DM-domain containing transcription factor Doublesex1 (DSX1) is expressed spatiotemporally in male-specific traits and orchestrates male trait formation in both somatic and gonadal tissues. However, it remains unknown how the *dsx1* gene is silenced in females to avoid male trait development. Heterochromatin Protein 1 (HP1) plays a crucial role in epigenetic gene silencing during developmental processes. Here we report the identification of four *HP1* orthologs in *D. magna*. None of these orthologs exhibited sexually dimorphic expression, and among them, *HP1-1* was most abundantly expressed during embryogenesis. The knock-down of *HP1-1* in female embryos led to the derepression of *dsx1* in the male-specific traits, resulting in the development of male characteristics, such as the elongation of the first antennae. These results suggest that HP1-1 silences *dsx1* for female development while environmental cues unlock this silencing to induce male production. We infer the HP1-dependent formation of a sex-specific chromatin structure on the *dsx1* locus is a key process in the environmental sex determination of *D. magna*.

## 1. Introduction

Sex determination is a fundamental biological process that governs the differentiation of somatic tissues and gonads, leading to sex-specific differences in physiology and behavior. Broadly, sex determination can be classified into two categories: genetic sex determination (GSD) and environmental sex determination (ESD) [1]. In GSD, sex-specific developmental pathways are activated by genes that segregate according to chromosomal inheritance, frequently involving sex chromosomes. In contrast, ESD relies on external environmental factors to initiate alternative regulatory cascades that govern the expression of male- or female-specific sex-determining genes. GSD mechanisms have been extensively studied in model organisms such as nematodes, fruit flies, and mice [1]. ESD is also widespread across animal taxa and exhibits diverse modes depending on the type of environmental cue and the timing of the sex determination [2]. DM-domain transcription factors, originally identified in *Drosophila melanogaster*, form a conserved family of regulators that play key roles in the GSD system [3]. In insects, the *doublesex* (*dsx*) gene regulates somatic sexual differentiation [4], while in vertebrates, *dmrt1* is essential for testis development in species such as birds [5] and mammals [6]. Subsequently, the DM-domain transcription factors have also been reported as key regulators of sexual differentiation in animals with ESD, such as the red-eared slider turtle *Trachemys scripta* [7] and the freshwater crustacean *Daphnia magna* [8]. These findings suggest a deep conservation of sex-determining mechanisms between GSD and ESD systems at the genetic level.

*D. magna* produces females by parthenogenesis in a healthy population. In this mode of reproduction, females produce diploid eggs without fertilization, which develop into clonal daughters. However, in response to environmental stressors, such as crowding and/or a shortening photoperiod, it produces males [9]. These males are clonal siblings of the females, as both are derived from unfertilized eggs produced by the same mother. Thus, sex determination in *D. magna* occurs in the absence of genetic differences between males and females. The environmental cues are detected by adult females, and transduced into juvenile hormone (JH) signaling, likely via a neuroendocrine system, leading to the production of oocytes that are destined to develop as males after ovulation [10]. During the development of male-destined eggs, the DM-domain containing transcription factor Doublesex1 (DSX1) is expressed spatiotemporally in male-specific traits, driving male differentiation and its maintenance [8]. This gene functions as a master regulator of male development, activating downstream pathways that lead to the formation of male-specific structures, such as the elongated first antenna and modified thoracic appendages. In contrast, *dsx1* is silenced in females throughout their lifespan, as demonstrated by a knock-in reporter study that showed no detectable expression in females from embryonic stages to adulthood, supporting the idea that environmental cues unlock the silencing of this male-determining gene [11].

Heterochromatin Protein 1 (HP1) is a highly conserved chromosomal protein found in fungi, plants, and animals [12]. It plays a crucial role in epigenetic gene silencing during development. HP1 mediates transcriptional repression by recognizing histone H3 lysine 9 methylation (H3K9me), promoting heterochromatin formation and gene silencing [13]. This function has been demonstrated across diverse species, including *D. melanogaster* [14], mammals [15], and *Arabidopsis thaliana* [16], highlighting its conserved function in gene regulation. Canonical HP1 consists of an N-terminal chromodomain (CD) and a C-terminal chromoshadow domain (CSD) connected by a hinge region [12]. In addition, non-canonical HP1 variants containing only a CD or CSD have been reported in *D. melanogaster* [17]. HP1 has been shown to play sex-specific roles in *Caenorhabditis elegans* [18], *D. melanogaster* [19], and mice [20], all of which employ the GSD system. In *C. elegans*, the HP1 homolog HPL-2 regulates male development [18]. In *D. melanogaster*, sex-specific distribution of HP1a suggests its role in sexual development [19]. In male mice, HP1γ is essential for spermatogenesis and germ cell survival [20]. However, its role in ESD remains largely unknown. Given that *dsx1* remains stably silenced in females throughout life, and that HP1 proteins mediate heritable gene repression through heterochromatin formation, we hypothesized that HP1 might contribute to maintaining the silenced state of *dsx1* in female *D. magna*. In this study, we identified an HP1 ortholog, HP1-1, that silences the *dsx1* gene in *D. magna* females.

## 2. Materials and Methods

### 2.1. Daphnia Strain and Transgenic Lines Culture

All *D. magna* wild-type and transgenic lines share the same genetic background (NIES strain) and were cultured in ADaM medium as previously described [21]. The *dsx1*-reporter strain expresses the mCherry under the control of the *dsx1* promoter/enhancer [11]. This line also expresses eGFP fused to the histone H2B (H2B-eGFP) gene under the control of the *elongation factor 1α1* (*EF1α1*) promoter/enhancer.

### 2.2. Hormone Treatment

Male daphniids were obtained by exposing 2- to 3-week-old females to 1 μg/L of the synthetic JH analog Fenoxycarb (Wako Pure Chemical, Osaka, Japan) for 16 h [22]. To ensure the accurate induction of male embryos, we selected maternal individuals approximately 10 h prior to the known critical period of sex determination (52–56 h after the previous oviposition) [23,24]. These individuals were then exposed to Fenoxycarb, which reliably induces male production under this condition. To confirm the uniformity of sex determination, we verified the sex of the offspring from both the first and last mothers used for embryo collection. In all experiments, only batches in which all offspring from these two mothers were confirmed to be male were included in downstream analyses. This protocol ensures consistent and effective male induction while accounting for potential variation in the hormonal response.

### 2.3. Identification and Alignment of HP1 Orthologs

The genomic locations of each *HP1* ortholog were examined using *tblastn* searches against the *D. magna* genome assembly ASM2063170v1.1 from the NCBI database (http://www.ncbi.nlm.nih.gov/ (accessed on 1 July 2022)). Amino acid sequences of HP1 orthologs from *D. melanogaster* (Appendix A) were obtained from the NCBI database and used as queries. To color the alignment, we used the Color Align Conversion tool from the Sequence Manipulation Suite, a freely available collection of JavaScript programs [25].

### 2.4. Quantitative RT-PCR

To analyze the expression levels of *HP1* orthologs during embryogenesis in *D. magna*, male and female embryos were collected at different time points: 0, 12, 24, 48, and 72 h after ovulation. These samples were subjected to RT-qPCR using cDNA synthesized from total RNA extracted from 20 embryos at each stage. In brief, total RNA was extracted in triplicate using Sepasol-RNA I Super G solution (Nacalai Tesque, Kyoto, Japan). cDNA synthesis was performed using the PrimeScript II 1st strand cDNA Synthesis Kit (Takara Bio, Shiga, Japan) with random primers. The real-time qPCR assay was conducted using Power SYBR Green PCR Master Mix (Applied Biosystems, Foster City, CA, USA) and gene-specific primers designed to amplify short PCR products (<200 bp) on a StepOnePlus Real-Time PCR system (Applied Biosystems) under the following conditions: 95 °C for 10 min, followed by 40 cycles of 95 °C for 15 s and 60 °C for 1 min. Relative expression levels were calculated using the Ct values, with the *ribosomal protein L32* gene serving as the reference control.

To measure the expression levels of *HP1-1* and *dsx1* in RNAi and overexpression experiments, injected embryos were collected. For total RNA purification, two or three embryos were used per each sample or replicate. Total RNA was extracted in triplicates as described above except that 10 μg of yeast tRNA (Thermo Fisher Scientific, Waltham, MA, USA) was added to each sample as a carrier. cDNA synthesis was performed using the PrimeScript II 1st strand cDNA Synthesis Kit (Takara Bio, Shiga, Japan) with random primers and subjected to RT-qPCR analysis. Gel electrophoresis and dissociation curve analysis were performed to confirm the correct amplicon size with the absence of nonspecific bands. Primer sequences for RT-qPCR are provided in Appendix A.

### 2.5. Microinjection

Following the protocol for microinjection [26], eggs were obtained from 2- to 3-week-old *D. magna* just after ovulation and transferred to ice-chilled M4 medium [27] containing 80 mM sucrose (M4-sucrose). For each experiment, 2 mM of Lucifer Yellow dye (Invitrogen, Carlsbad, CA, USA) was added to the injection solution as a visible maker. Microinjections were performed on ice. The surviving embryos were transferred individually to each well of a 96-well plate containing 100 μL of M4-sucrose and then incubated for 96 h at 23 °C without any change of the medium.

### 2.6. HP1-1 RNAi

The *HP1-1* CDS was used to design *HP1-1*-targeting siRNA with the Block-iT RNAi Designer (https://rnaidesigner.thermofisher.com/rnaiexpress/ (accessed on 8 January 2019)). An siRNA targeting the *E. coli malE* gene was also designed and used as a control (Ctrl) siRNA. The siRNA sequences are shown in Appendix A. Two dTdT nucleotides were added to the 3′ end of each siRNA strand. To assess the specificity of siRNAs, we aligned the siRNA sequences against the three non-target HP1 paralogs identified in this study. Both siRNAs targeting HP1-1 showed at least five mismatches with the non-target paralogs, including multiple mismatches within the seed region (positions 2–8) (Appendix A). Given the critical role of the seed region in RISC-mediated recognition, this mismatch pattern suggests a low likelihood of off-target silencing. This approach is consistent with established principles of RNAi design [28]. The siRNA oligonucleotides were dissolved in DNase/RNase-free water (Life Technologies, Grand Island, NY, USA). For microinjection, 100 μM of each siRNA was used. For phenotypic and fluorescence analysis, embryos were collected at 48 h post-injection. This time point was chosen based on time-course observations showing that Dsx1-mCherry expression becomes clearly detectable in male-specific traits, particularly the elongation of the first antenna. The 48 h stage represents a point at which both morphological and molecular markers of male development are reliably visible, enabling robust assessment of RNAi effects.

### 2.7. Ectopic Expression of HP1-1

We amplified the DNA fragment of the *HP1-1* CDS by PCR using KOD-plus (Toyobo, Osaka, Japan) as the DNA polymerase and cDNA obtained from *D. magna* embryos. The primer sequences were as follows: forward, 5′-ATGGGTCGAAGCACAAAAG-3′ and reverse, 5′-GTCCGTTCCATCTTTCTCCT-3′. The expected amplicon size was 618 bp. The PCR product was purified and concentrated using a QIAGEN MinElute PCR purification Kit (Qiagen, Germantown, MD, USA). We constructed a bicistronic expression plasmid for *HP1-1* and *eGFP* expression under the control of *D. magna EF1α1* promoter/enhancer. The CDS of *H2B* in pCS-EF1α1::H2B-eGFP [29] was replaced by HP1-1-P2A using the GeneArt Seamless Cloning and Assembly Enzyme Mix (Invitrogen), resulting in pCS-EF1α1::HP1-1-P2A-eGFP. To generate the control plasmid, HP1-1-P2A was removed from pCS-EF1α1::HP1-1-P2A-GFP by the PCR-based method [30]. The 2A peptide allows for the expression of HP1-1 and eGFP from a single mRNA, with efficient co-translational cleavage and proper localization of proteins [31], which allowed us to confirm HP1-1 expression in embryos. Among the 2A peptides, the P2A peptide, derived from porcine teschovirus-1, has emerged as a highly efficient tool for bicistronic and polycistronic gene expression in eukaryotic systems [32]. The bicistronic expression plasmid and control plasmid were named pHP1 and pGFP, respectively, in this study. Each plasmid was injected into eggs of the *dsx1*-reporter strain destined to become males. The plasmid concentration was 50 or 100 ng/µL. Injected eggs were cultured until 30 h after injection to observe and calculate the fluorescence intensity differences. This timing corresponds to the early developmental stage when sex-specific differences in *dsx1* expression begin to begin to be observed clearly. Given that *HP1-1* overexpression was initiated at the one-cell stage and is transient, the 30 h time point was chosen to capture its regulatory effect on *dsx1* before *HP1-1* expression declines in later stages and its impact is no longer detectable. Three independent replicates, each containing three embryos, were collected at 30 h after injection and subjected to total RNA isolation and cDNA synthesis as described above.

### 2.8. Quantitation of the Fluorescence

Observations were performed, and photos were taken with a Leica DC500 CCD Digital Camera equipped with a Leica M165FC fluorescence microscope (Leica Microsystem, Mannheim, Germany). Fluorescence images were obtained using GFP and mCherry filters under the following conditions: 1.0 s exposure time, 3.0× gain, 1.0 saturation and 1.0 gamma for GFP, and 2.0 s exposure time, 8.0× gain, 1.0 saturation, and 1.6 gamma for mCherry. mCherry and GFP fluorescence intensities were calculated using ImageJ software (version 2.16.0/1.54p) [33]. A region of interest (ROI) was drawn around each embryonic structure to measure total pixel intensity and area. To account for background fluorescence, three smaller ROIs were drawn near the embryo’s structure, and their mean pixel intensities were calculated. The corrected fluorescence intensity was obtained by subtracting the product of the background’s mean pixel intensities and the embryo’s ROI area from the embryo’s total pixel intensity. In addition, relative fluorescence intensity (RFI) was calculated by dividing the corrected fluorescence intensity of the injected embryos with those of the uninjected embryos from the same clutch to minimize differences in autofluorescence between embryos from different mothers. The RFIs of the control samples were compared to the RFIs of the treated embryos. At least 6 embryos were used to quantify fluorescence.

## 3. Results

### 3.1. Four HP1 Orthologs on the D. magna Genome

We searched the *D. magna* reference genome (GenBank accession no. GCF_020631705.1) for sequences orthologous to *D. melanogaster HP1*. We identified three canonical HP1 orthologs, each containing the CD and CSD connected by a hinge region. These were designated as *HP1-1*, *HP1-2*, and *HP1-3*, respectively (Figure 1, Appendix A). The predicted amino acid sequences of the CDs and CSDs from the three *HP1* orthologs exhibited a high similarity with those of *Drosophila* and human (Appendix A). Functionally important residues involving in the aromatic cage for H3K9me3 recognition in CD [34] and in the dimerization surface in the CSD [35] are conserved in these HP1 orthologs, confirming the presence of four *HP1* orthologs in the *D. magna* genome. In addition, we identified a noncanonical HP1 variant, HP1cd, which lacks a CSD but contains three CDs (Figure 1, Appendix A). This CD-only-*HP1*, *HP1cd*, was tandemly located with *HP1-2* on LG7 (Appendix A).

### 3.2. HP1-1 Is the Most Abundantly Expressed During Embryogenesis Among HP1 Orthologs

We examined the expression levels of the four *HP1* orthologs during embryogenesis by quantitative RT-PCR. None of the *HP1* orthologs exhibited sex-specific differences in expression, including *HP1-1* at 48 h, where variation among replicates was observed (Figure 2). Among them, *HP1-1* exhibited the highest expression levels throughout development, with a peak immediately after ovulation (0 h post-ovulation, 0 hpo), followed by a gradual decline over time. Similarly, *HP1-3* showed elevated expression during early embryogenesis (0 and 12 hpo) with a decreasing trend in later stages. In contrast, *HP1-2* and its adjacent gene *HP1cd* maintained consistently low expression levels, except at 0 hpo. These results identify *HP1-1* as the predominantly expressed *HP1* gene during embryogenesis, leading us to prioritize *HP1-1* for further functional analysis.

### 3.3. HP1-1 Silencing Derepressed Dsx1 and Led to Masculinization in Female Embryos

To elucidate the role of *HP1-1* in ESD, we knocked down *HP1-1* expression by RNA interference (RNAi) during embryogenesis. To assess the effect of *HP1-1* knockdown on *dsx1* expression, we utilized the *dsx1*-reporter line that visualizes *dsx1* expression via mCherry fluorescence. This transgenic line expresses mCherry under the control of *dsx1* promoter/enhancer, along with nucleus-localized eGFP driven by the ubiquitously active *EF1α1* promoter/enhancer [11].

We injected 100 μM of *HP1-1* siRNA, designated siHP1-1a, into eggs destined to develop as females. We analyzed these individuals at both embryonic and juvenile stages to assess the impact of *HP1-1* knockdown on *dsx1* expression and sexual development. At stage 12 (48 h after injection) [36], male-specific mCherry fluorescence was observed in male traits such as first antennae (An1) and first thoracic appendage (T1) in control males and absent in siCtrl females (Figure 3A). In contrast, siHP1-1a-injected female embryos exhibited increased mCherry fluorescence throughout the body, with particularly strong signals in male-specific structures (Figure 3A). At the juvenile stage, *HP1-1* knockdown led to elongation of the first antennae and stronger mCherry fluorescence (Figure 3B, Appendix A). Despite reaching the juvenile stage, all siHP1-1a–injected individuals became immobile and subsequently died before clear gonadal development could occur (Table 1). As a result, it was not possible to determine whether any sex-specific differentiation of the gonads had taken place.

The relative fluorescence intensity (RFI) of mCherry, which reflects *dsx1* activity, significantly increased in siHP1-1a–injected embryos compared to siCtrl, indicating derepression of *dsx1* (Figure 4A). Quantitative RT-PCR showed that *HP1-1* expression was reduced to 9.3% of siCtrl embryos, while *dsx1* expression was upregulated up to 15-fold compared to siCtrl embryos (Figure 4B). We confirmed the specificity of the *HP1-1* knockdown by using an independent siRNA, siHP1-1b (Table 1, Figure 3 and Figure 4).

We also examined the effects of *HP1-1* RNAi on male embryos. Similar to siHP1 female embryos, *HP1-1* depletion did not produce viable, swimming juveniles (Appendix A). However, *HP1-1* knockdown did not affect *dsx1* expression in male embryos (Appendix A). Taken together, these findings suggest that *HP1-1* represses *dsx1* expression in females.

### 3.4. HP1-1 Overexpression Alone Did Not Change Dsx1 Expression in Male Embryos

We investigated whether *HP1-1* overexpression leads to the silencing of *dsx1* expression in males. We constructed a plasmid expressing *HP1-1* under the control of ubiquitously active *EF1α1* promoter/enhancer [29]. We injected two different concentrations of the plasmid (50 ng/µL or 100 ng/µL) into eggs of the *dsx1-*reporter line, which were destined to develop as males. We examined the phenotypes of the injected embryos at stage 7 (30 h after injection) when *dsx1* shows significant sexually dimorphic expression [8,11]. In embryos injected with 50 ng/µL of the *HP1-1* overexpression plasmid (pHP1), mCherry fluorescence intensity and *dsx1* transcript levels did not differ significantly from those in control embryos injected with the *GFP* plasmid (pGFP), as shown in Figure 5A–C. *HP1-1* transcript levels were significantly elevated by approximately 2.7-fold in pHP1-injected embryos, confirming successful overexpression (Figure 5C). The injection of 100 ng/µL *HP1-1* plasmid resulted in early embryonic lethality, preventing further analysis of *Dsx1* expression (Appendix A). These findings suggest that *HP1-1* overexpression alone is insufficient to suppress *dsx1* expression in males.

## 4. Discussion

Unlike genetic sex determination (GSD), where chromosomal or genetic factors initiate sex-specific development, environmental sex determination (ESD) systems rely on external cues such as temperature, photoperiod, or population density. Although ESD occurs across diverse taxa, the molecular mechanisms linking environmental signals to sex-specific gene regulation remain poorly understood. In *D. magna*, we previously demonstrated that the male-determining gene *dsx1* is silenced in females and becomes derepressed in response to environmental stimuli, thereby functioning as a master regulator of male development [8]. In this study, we identified four *HP1* orthologs and analyzed the function of one—*HP1-1*—during embryogenesis. Our findings demonstrate that *HP1-1* silences *Dsx1* expression to promote female development.

### 4.1. Conserved Structure and Sex-Specific Function of HP1-1

HP1-1 in *D. magna* possesses conserved features that are critical for its role in chromatin-mediated gene regulation. Specifically, residues essential for forming the aromatic cage that recognizes H3K9me3 in the CD [34], as well as residues involved in dimerization through the CSD [35], are well conserved. These structural elements suggest that HP1-1 is capable of interacting with methylated histones and forming dimers with other chromatin-associated proteins, supporting its proposed function in heterochromatin formation and gene silencing.

*HP1-1* did not show sexual dimorphism in gene expression, although it showed a transient difference in expression between male and female embryos at 48 h. This variation fell within the range of biological variability and was not statistically significant. In other organisms, such as *C. elegans*, *D. melanogaster*, and mice, sexually dimorphic expression patterns of HP1 family proteins have not been consistently reported despite their sex-specific functions [18,19,20]. These findings collectively suggest that the sex-specific role of *HP1-1* in *D. magna* is not governed by its expression level but rather by its chromatin localization, interaction with other regulatory proteins, or its recruitment to specific genomic loci.

Taken together, our results support the idea that HP1-1 functions as a conserved chromatin modifier whose sex-specific regulatory role arises through context-dependent mechanisms rather than expression asymmetry. This highlights a broader principle in epigenetic regulation, whereby conserved structural elements enable plastic yet specific developmental outcomes depending on the cellular or environmental context.

### 4.2. Environmental Sex-Determining Mechanisms in D. magna

*dsx1* expression is activated in males via juvenile hormone signaling and the transcription factor Vrille in response to environmental cues. *dsx1* remains active throughout life [37] (Figure 6). In contrast, during female development, *dsx1* must be silenced to prevent masculinization. Notably, silencing a single gene, *HP1-1*, led to *dsx1* derepression in females. In *D. melanogaster*, spatial and temporal *dsx* expression is regulated by *Hox* genes [38] and hormone signaling [39]. Similarly, in *D. magna*, non-sexually dimorphic *dsx1* transcriptional regulators may exist in females, but HP1-1-dependent epigenetic silencing likely shields the *dsx1* locus from their influence.

Building on the findings in this study, we propose a model in which HP1-1 establishes a heritable silent chromatin environment at the *dsx1* locus in females, thereby preventing the initiation of male development (Figure 6). This transcriptional repression is reinforced post-transcriptionally by SHEP, which blocks the leaky translation of any residual Dsx1 transcripts [40]. Together, HP1-1 and SHEP ensure the robust suppression of *dsx1* in the female developmental pathway. In contrast, in male embryos, the *dsx1* transcripts are translated with the aid of the long noncoding RNA *DAPALR*, which sequesters the translational repressor SHEP [41]. This dual activation—transcriptional and translational—enables *dsx1* expression to drive male-specific development. Future research should aim to dissect how HP1-1–dependent silencing is lifted in males, and how these multiple layers of transcriptional and translational regulation are integrated and temporally coordinated during sex-specific differentiation.

### 4.3. Epigenetic Regulation for Sex Determination

In many animals, sex-specific development is directed by a combination of activating male or female programs and repressing the alternative fate. In GSD systems, the activation of male programs is controlled by chromosomal or genetic cues—for example, *sry* in mammals [42] or *dmrt1* in birds [5]. In addition, the active repression of the opposing sexual pathway is often essential [43]. ESD systems, such as that of *D. magna*, rely on environmental signals like photoperiod or population density to initiate sex-specific development. Our current findings demonstrate that the male developmental pathway in *D. magna* is repressed by HP1-1, a heterochromatin-associated factor. In reptiles with temperature-dependent sex determination, epigenetic mechanisms are also crucial for activating *DMRT1*, a testis-determining gene belonging to the DM-domain transcription factor family, which includes *Dsx1* [44]. Thus, while GSD and ESD differ in the initiating cues (genetic vs. environmental), both systems may depend on epigenetic mechanisms to balance the activation and repression of sex-specific developmental programs.

We acknowledge that H3K9me3 levels were not directly assessed in our current study. Notably, H3K9me3 has been previously detected in *D. magna* male gonads using both immunofluorescence and western blotting [45]. While our study focused on embryonic stages, future work should explore H3K9me3 distribution at the *dsx1* locus in both male and female embryos to test this hypothesis more directly.

In conclusion, we demonstrate that HP1-1 silences the male sex-determining *dsx1* gene to promote female development in *D. magna*. This finding underscores the role of epigenetic mechanisms in ESD pathways and offers new insights into the diversity and evolution of the sex-determination systems.

## Figures and Tables

**Figure 1 jdb-13-00023-f001:**
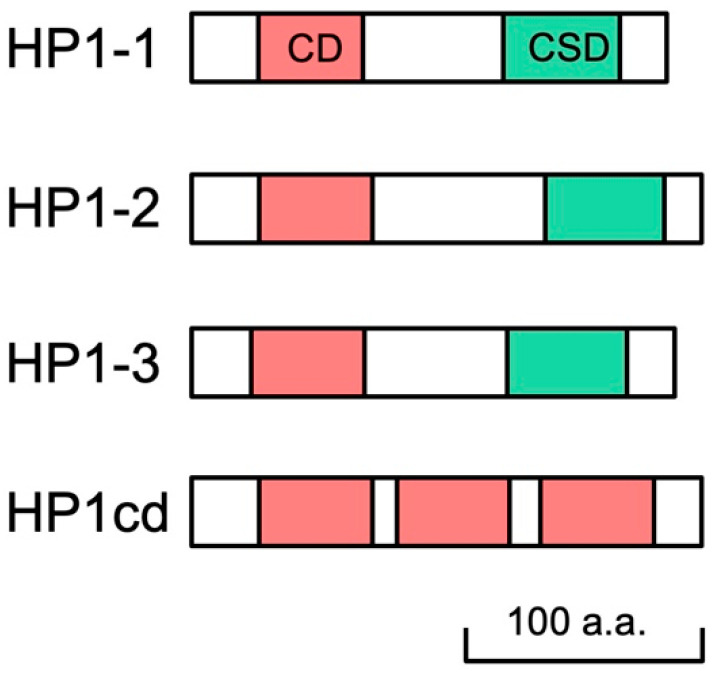
Schematic representation of four HP1 orthologs in *D. magna*. *Daphnia* HP1 proteins contain both chromodomain (CD) and chromoshadow domain (CSD) in their sequences, except HP1cd containing 3 CDs only.

**Figure 2 jdb-13-00023-f002:**
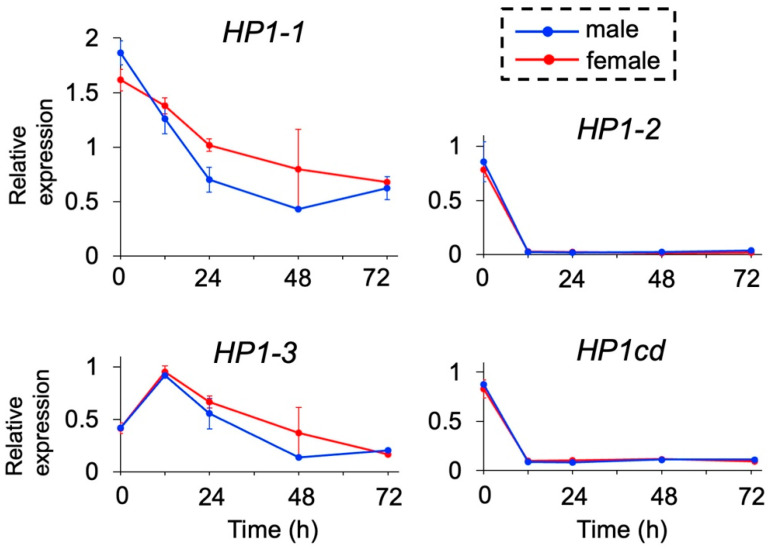
Temporal expression of *HP1* orthologs in *D. magna*. Male and female embryos were obtained, and gene expression levels were analyzed at 0, 12, 24, 48, and 72 h after ovulation by quantitative RT-PCR. Twenty individuals were included in each replicate. Error bars indicate S.E.M (*n* = 3).

**Figure 3 jdb-13-00023-f003:**
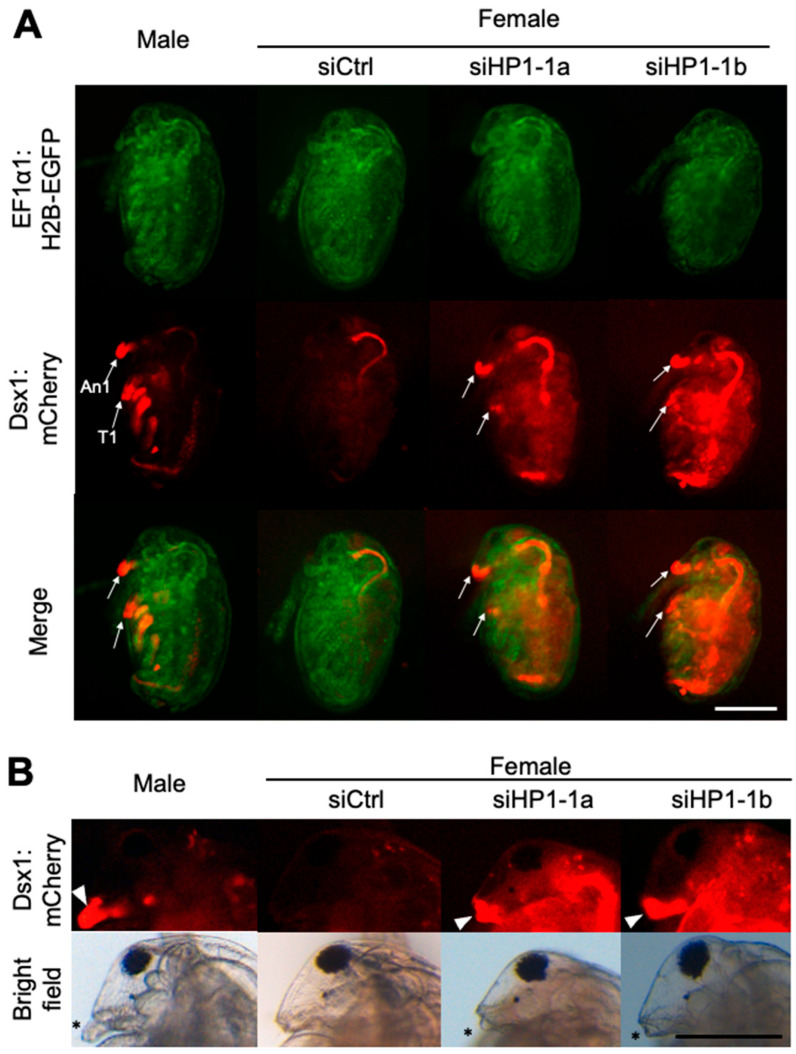
Phenotypes and gene expressions of *HP1-1* RNAi females. (**A**) Phenotypes of female embryos injected with *HP1-1* siRNAs. Embryos from the *dsx1* reporter line were injected with either control siRNA (siCtrl) or *HP1-1* siRNAs (siHP1-1a, siHP1-1b) just after ovulation and observed at stage 12 (48 h post-injection). Fluorescence from EF1α1:H2B-EGFP (green) and Dsx1:mCherry (red) is shown. An1: first antenna, T1: first thoracic appendage. The scale bar represents 200 µm. (**B**) Phenotypes of juveniles derived from embryos injected with *HP1-1* siRNAs. A male juvenile from the *dsx1* reporter line is included for comparison (leftmost panel). The white arrowhead indicates Dsx1:mCherry signal at the base of the first antenna. Asterisks mark the position of the first antenna. The scale bar represents 200 µm.

**Figure 4 jdb-13-00023-f004:**
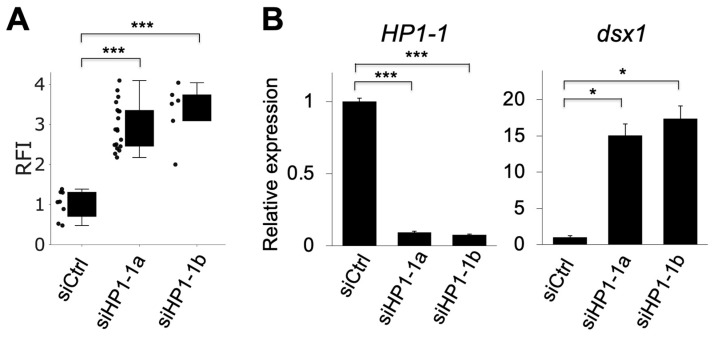
Quantitative analysis of mCherry fluorescence and gene expression following *HP1-1* knockdown. (**A**) Relative fluorescence intensity (RFI) of mCherry in dsx1-reporter female embryos injected with control siRNA (siCtrl), *HP1-1* siRNA-1 (siHP1-1a), or *HP1-1* siRNA-2 (siHP1-1b). Each dot represents one embryo. Asterisks indicate statistical significance (*** *p* < 0.001, *t*-test). (**B**) Quantitative RT-PCR analysis of *HP1-1* (left) and *dsx1* (right) transcript levels in embryos injected with siCtrl, siHP1-1a, or siHP1-1b. Expression levels were normalized to *ribosomal protein L32*. Error bars indicate S.E.M (*n* = 3). * *p* < 0.05, *** *p* < 0.001 (Student’s *t*-test).

**Figure 5 jdb-13-00023-f005:**
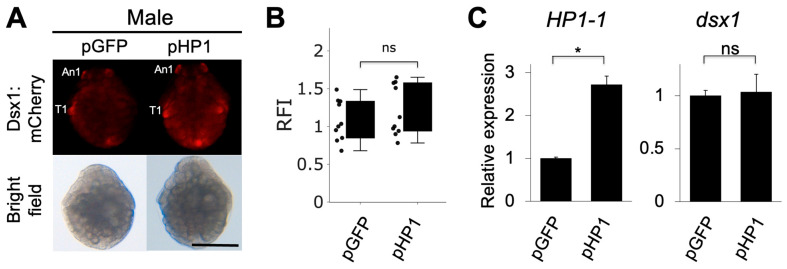
Phenotypes and gene expressions of males injected with *HP1-1* expression plasmids. (**A**) Phenotypes of male embryos injected with the *HP1-1* expression plasmid (pHP1). The *GFP* expression plasmid (pGFP) was used as a control. Embryos from the *dsx1* reporter line were injected just after ovulation and observed at stage 7 (30 h after injection). The scale bar represents 200 µm. (**B**) Relative fluorescence intensity (RFI) of pHP1-injected embryos, normalized to pGFP-injected controls. (**C**) Quantitative RT-PCR analysis of *HP1-1* (left) and *dsx1* (right) transcript levels in embryos injected with each plasmid. RT-qPCR results are shown as expression levels relative to pGFP embryos. Error bars indicate S.E.M (*n* = 3). * *p* < 0.05, ns: not significant (Student’s *t*-test).

**Figure 6 jdb-13-00023-f006:**
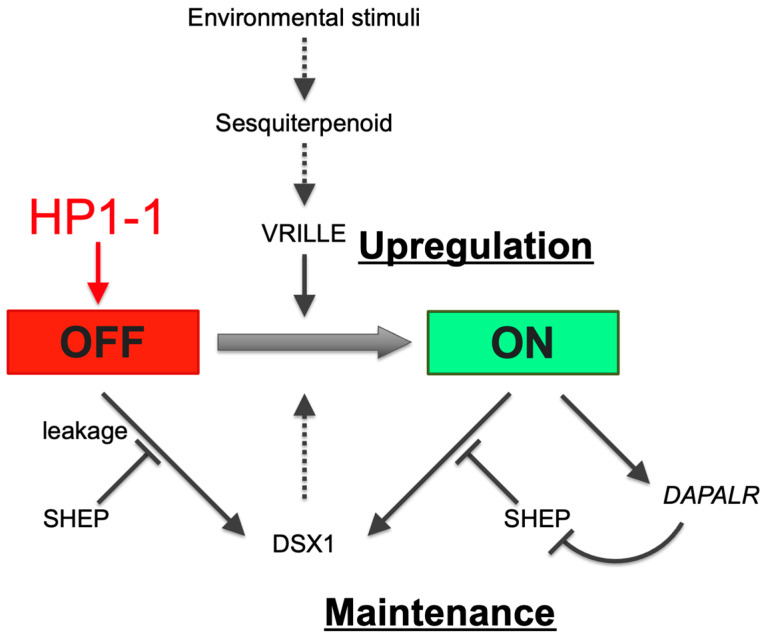
HP1-1–dependent repression of *dsx1* in females via heterochromatin formation. In response to environmental stimuli, juvenile hormone or sesquiterpenoid hormone signaling activates the bZIP transcription factor gene *vrille*, leading to *dsx1* transcription. *dsx1* mRNA is further regulated at the translational level: SHEP represses leaky translation in females, while in males, the lncRNA *DAPALR* inhibits SHEP to permit *dsx1* protein synthesis. Black arrows indicate activation, black bars indicate repression, and the red arrow from HP1-1 to “OFF” denotes silencing via heterochromatin formation. Solid and dotted lines indicate validated and non-validated relationships by experiments. The red and green background colors of the OFF and ON boxes represent repressed and active transcriptional states, respectively. Adapted and modified from Kato and Watanabe (2022) [10].

**Table 1 jdb-13-00023-t001:** Effects of *HP1-1* RNAi on female embryos of the *Dsx1*-reporter strain.

siRNA	Injected Embryo	Hatched Embryo	Survived Juvenile	Immobile Juvenile	Increase of mCherry Fluorescence
siCtrl	9	9	8	0	0% (0/8)
siHP1-1a	23	22	18	18	100% (18/18)
siHP1-1b	10	10	6	6	100% (6/6)

## Data Availability

The dataset analyzed during the current study is available in the NCBI repository, GCF_020631705.1 to the *D. magna* reference genome.

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
