# Peer review of "HP1-Mediated Silencing of the Doublesex1 Gene for Female Determination in the Crustacean Daphnia magna"

_jdb, 2025, doi:10.3390/jdb13030023_

Round 1
Reviewer 1 Report
Comments and Suggestions for Authors
The manuscript addresses a compelling and timely topic in the field of developmental and evolutionary biology: the epigenetic silencing of the doublesex1 gene in female Daphnia magna as a mechanism to prevent male trait formation. The study is well-executed and builds on the authors’ strong expertise in Daphnia developmental biology and gene regulation. Overall, the manuscript is clearly written, and the conclusions are generally well supported by the data presented.
That said, a few minor issues should be addressed prior to publication to improve the manuscript’s clarity and impact:
GSD vs. ESD
The concepts of genetic sex determination (GSD) and environmental sex determination (ESD) are briefly introduced but not discussed further in the context of the findings. It would strengthen the manuscript to reflect on how HP1-mediated silencing of dsx1 fits into the broader framework of ESD in Daphnia, particularly in comparison to GSD systems.
Contextualization of Male Development and HP1
The introduction could benefit from a more accessible explanation of male developmental pathways in Daphnia, especially for readers unfamiliar with the system. In particular, positioning HP1 more clearly within this regulatory framework would help clarify why it was chosen as a candidate of interest and how it is hypothesized to influence dsx1 silencing.
Clarification of “Male-destined” and “Female-destined” Embryos
The manuscript frequently refers to “male-destined” and “female-destined” embryos based on MF treatment. However, it would be important to acknowledge the possibility of phenotypic variation—especially under suboptimal conditions—and clarify how the sex of embryos was verified post-treatment. Were all MF-treated individuals confirmed to be male?
Fluorescence Quantification
The methods section would benefit from more detail on the quantification of fluorescence signals. Specifically, were the reported values background-corrected, and how was this done? If this step was performed, it should be clearly described in the manuscript.
Specificity of RNAi Probes
While it is commendable that the authors used two independent RNAi constructs to mitigate off-target effects, it would be helpful to provide information on the sequence similarity between the four HP1 orthologs. Can off-target silencing of other HP1 paralogs be ruled out entirely?
Timing of Sample Collection Post-injection
The rationale behind sampling females 48h and males 30h post-injection is unclear. Please elaborate on the biological or experimental justification for this difference in timing.
Balance in Discussion
The discussion effectively interprets the main findings, but it currently focuses heavily on H3K9 methylation. A broader discussion that places the results into the overall developmental cascade of male determination in Daphnia—or into comparable epigenetic regulatory mechanisms across taxa—would improve balance and broaden the manuscript’s relevance.
Overall, this is a strong and promising manuscript. With these minor revisions, it will be even better positioned to make a meaningful contribution to our understanding of epigenetic control in environmentally determined sex systems.
Author Response
Reviewer 1
The manuscript addresses a compelling and timely topic in the field of developmental and evolutionary biology: the epigenetic silencing of the doublesex1 gene in female Daphnia magna as a mechanism to prevent male trait formation. The study is well-executed and builds on the authors’ strong expertise in Daphnia developmental biology and gene regulation. Overall, the manuscript is clearly written, and the conclusions are generally well supported by the data presented.
That said, a few minor issues should be addressed prior to publication to improve the manuscript’s clarity and impact:
Response: We thank Reviewer 1 for the positive evaluation and thoughtful feedback. We appreciate the recognition of our study’s relevance to epigenetic silencing and sex determination, and have addressed all points raised as follows.
GSD vs. ESD
The concepts of genetic sex determination (GSD) and environmental sex determination (ESD) are briefly introduced but not discussed further in the context of the findings. It would strengthen the manuscript to reflect on how HP1-mediated silencing of dsx1 fits into the broader framework of ESD in Daphnia, particularly in comparison to GSD systems.
Response (lines 379–391): Thank you for the thoughtful suggestion. We have revised the Discussion section to compare the HP1-1–mediated silencing of dsx1 in Daphnia magna with mechanisms used in GSD systems. While GSD often involves activation of sex-determining genes, repression of the opposite pathway also plays an essential role. By emphasizing this shared reliance on both activation and repression, we contextualize our findings within a broader regulatory logic common to both GSD and ESD systems.
Contextualization of Male Development and HP1
The introduction could benefit from a more accessible explanation of male developmental pathways in Daphnia, especially for readers unfamiliar with the system. In particular, positioning HP1 more clearly within this regulatory framework would help clarify why it was chosen as a candidate of interest and how it is hypothesized to influence dsx1 silencing.
Response (lines 81–84): We thank the reviewer for this helpful suggestion. We have revised the Introduction section to clarify the male developmental pathway in Daphnia magna, including the role of Dsx1 in specifying male traits. We have also added a rationale for selecting HP1 as a candidate epigenetic regulator, highlighting its known role in heterochromatin-mediated gene silencing and the potential for maintaining Dsx1 repression in females. Please see the revised Introduction for the updated text.
Clarification of “Male-destined” and “Female-destined” Embryos
The manuscript frequently refers to “male-destined” and “female-destined” embryos based on MF treatment. However, it would be important to acknowledge the possibility of phenotypic variation—especially under suboptimal conditions—and clarify how the sex of embryos was verified post-treatment. Were all MF-treated individuals confirmed to be male?
Response (lines 93–104): We appreciate the reviewer’s important point. In our study, we ensured accurate induction of male embryos by exposing maternal individuals to the Juvenile Hormone agonist Fenoxycarb approximately 10 hours prior to the established sex determination window (52–56 hours post-oviposition). To validate the treatment’s effectiveness, we confirmed that the offspring from both the earliest and latest treated mothers were all males. Only embryos derived from batches that passed this confirmation step were used in downstream experiments. We have now added a detailed description of this verification step in the Methods section.
Fluorescence Quantification
The methods section would benefit from more detail on the quantification of fluorescence signals. Specifically, were the reported values background-corrected, and how was this done? If this step was performed, it should be clearly described in the manuscript.
Response (lines 196–204): We appreciate the reviewer’s suggestion. To clarify, all fluorescence intensity measurements were indeed background-corrected. We have now revised the “Quantitation of the fluorescence” section in the Methods to explicitly describe the background correction procedure. Specifically, for each measurement, a region of interest (ROI) was drawn around the relevant embryonic structure to calculate total pixel intensity and area. To account for background fluorescence, three smaller ROIs were placed near the structure, and their mean pixel intensity was calculated. The corrected fluorescence was obtained by subtracting the product of the background intensity and the signal ROI area from the total signal intensity. This step was performed using ImageJ software. We hope this revision addresses the reviewer’s concern and provides greater clarity and reproducibility.
Specificity of RNAi Probes
While it is commendable that the authors used two independent RNAi constructs to mitigate off-target effects, it would be helpful to provide information on the sequence similarity between the four HP1 orthologs. Can off-target silencing of other HP1 paralogs be ruled out entirely?
Response (lines 148–154, Supplementary Figure S1): Thank you for this insightful comment. We agree that clarifying the specificity of the RNAi probes is important. To address this, we performed sequence alignments between the two siRNAs targeting HP1-1 and the three non-target HP1 paralogs. While short regions of partial similarity were observed, each siRNA contained at least five mismatches with the non-target paralogs, particularly within the seed region (positions 2–8 from the 5’ end), which is crucial for RNA-induced silencing complex (RISC) binding. These mismatches strongly reduce the risk of off-target silencing. We have added a sentence in the Methods section to clarify this point and cited Kamola et al. (2015), which demonstrates the importance of mismatch patterns outside the seed region in determining RNAi specificity.
Timing of Sample Collection Post-injection
The rationale behind sampling females 48h and males 30h post-injection is unclear. Please elaborate on the biological or experimental justification for this difference in timing.
Response (lines 156–161, lines 182–186): Thank you for this important question. We performed time-course observations for both male and female embryos and selected these time points based on key biological markers and technical feasibility. For females injected with siRNA, we selected 48 hours post-injection (stage 12) because this is when the sexually dimorphic elongation of the first antenna becomes clearly detectable if Dsx1 is de-repressed. Moreover, Dsx1-mCherry expression is robust and spatially localized at this stage, enabling reliable assessment of both morphological and molecular phenotypes. For males injected with HP1-1 expression plasmid, we selected 30 hours post-injection (stage 7). This timing corresponds to the early developmental stage when sex-specific differences in dsx1 expression begin to be observed clearly. Given that HP1-1 overexpression was initiated at the one-cell stage and is transient, the 30 h time point was chosen to capture its regulatory effect on dsx1 before HP1-1 expression declines in later stages and its impact is no longer detectable. We have clarified this rationale in the revised Materials and Methods section.
Balance in Discussion
The discussion effectively interprets the main findings, but it currently focuses heavily on H3K9 methylation. A broader discussion that places the results into the overall developmental cascade of male determination in Daphnia—or into comparable epigenetic regulatory mechanisms across taxa—would improve balance and broaden the manuscript’s relevance.
Response (lines 366–377, Figure 6): We appreciate the reviewer’s suggestion to broaden the scope of the discussion. In the revised manuscript, we have expanded the discussion to better integrate our findings into the overall developmental cascade of male determination in Daphnia magna. Specifically, we have positioned HP1-1 upstream in the regulatory hierarchy, acting to establish heritable silencing of Dsx1 in females, while highlighting how Vrille- and DAPALR-dependent mechanisms may antagonize this silencing in males. Furthermore, we clarified the role of HP1-1 in relation to Shep, proposing a two-layered repression model involving both transcriptional and translational control. In addition, we placed our findings in a broader epigenetic context across taxa. In the section titled “Epigenetic regulation for environmental sex determination,” we now compare the HP1-1–mediated silencing in Daphnia with repression mechanisms seen in GSD systems, such as mammals and birds, where repression of alternative sexual pathways is also essential. These revisions address both suggested perspectives and enhance the general relevance of our findings.
Overall, this is a strong and promising manuscript. With these minor revisions, it will be even better positioned to make a meaningful contribution to our understanding of epigenetic control in environmentally determined sex systems.
Response: We thank the reviewer for the positive assessment and encouraging remarks. In response to all the comments and suggestions, we have carefully revised the manuscript to improve clarity, contextual integration, and relevance across taxa. We believe these changes have strengthened the manuscript and better highlight its contribution to the field of epigenetic regulation in environmentally determined sex systems.
Reviewer 2 Report
Comments and Suggestions for Authors
I appreciate that the editor has consider my participation as reviewer in this manuscript.
This manuscript presents a compelling study suggesting that environmental cues trigger male production in Daphnia magna by lifting HP1-1-mediated silencing of the male-determining gene Dsx1. Knockdown of HP1-1 in females leads to the emergence of male traits, revealing its role in repressing male development under normal conditions.
Overall, this is a very interesting work, and is generally well designed and executed, however, I have some comments. Since it is difficult to give specific details as the MS is sent already formatted without line numbers, I will give the comments for each block.
Introduction:
The introduction is too brief, particularly regarding aspects of Daphnia biology and sex determination, which may not be familiar to all readers.
- Please expand on the sentence beginning “DM-domain transcription factor was first identified...” (line 8 of the introduction). I assume the authors are referring to the DM transcription factor family. Additional background on this family and its role in sex determination across species would help readers better contextualize the study.
- A clearer explanation of sex biology in Daphnia species is needed. Specifically, please clarify that males are genetically identical clones of their mothers (page 2, line 5). While this may seem obvious to Daphnia researchers, it is not for a broader audience.
- The sentence, “In contrast, Dsx1 is silenced in females throughout their lifespan, suggesting that environmental cues unlock the silencing of this male-determining gene,” needs a citation. Otherwise, this claim should be removed or rephrased. Also, if there is any work that has measure the expression levels of Dsw1 in juvenils versus adults, it should also be cited.
- The sentence “It plays a crucial role in epigenetic gene silencing during development [9–11].” needs more explanation and detail about the roles of HP1. Please elaborate on HP1’s known roles and specify the species used in the cited studies.
Materials and methods
The manuscript does not explain how the authors determined which eggs were “destined to develop as females” or “as males.” This phrasing appears several times and needs clarification.
From the current text, it appears that experiments were conducted blindly, and phenotypic sex was determined afterward. If this is the case, please clarify the environmental or experimental conditions (e.g., photoperiod, crowding, nutrient deprivation) under which males were induced.
If a specific male-inducing protocol was used—such as exposure to Methyl Farnesoate (MF)—this must be stated explicitly. Please include references (e.g., Olmstead & LeBlanc, 2002) and describe the details of the protocol, including whether it achieved 100% male offspring, to support reproducibility.
Results
Figure 1 and Table 1: It is unclear whether the siHP1-1a individuals survived the treatment. Table 1 notes they did not swim, but does this mean they died immediately, or were they alive but immobile? Were they able to reach a juvenile stage? Any additional observations—especially regarding gonadal development—would help clarify this point.
Figure 2: Supplementary Figure 2 contains important data and should be moved into the main figure set to improve accessibility and understanding.
Figures 3 and 4: Please explain the rationale for analyzing females at 48h post-injection (stage 12) and males at 30h (stage 7). If these time points differ for a biological reason, this should be clearly justified. Otherwise, a standardized time point would have been the ideal experimental procedure to improve comparability.
Figure 3 Legend: There is a typographical error. It currently reads: “The scale bar れ…” Please correct this.
Discussion
Please, homogenize the format. Introduction uses the terms GSD and ESD, however, they are not used in the discussion.
Although the conclusions of the work are very interesting, some of the discussion seems speculative.
For example, authors claim:
- “One possible explanation for this discrepancy is that transcriptional activation of Dsx1 in male-destined embryos may inhibit H3K9 trimethylation, preventing HP1-1 from binding to the Dsx1 locus.”,
However, this not directly supported by experimental data in the present MS. There is no evidence presented that H3K9me3 levels were examined in the various experimental groups (both in females and males).
This is particularly relevant given that this histone modification (H3K9m3) has previously been observed in Daphnia magna, specifically in male gonadas (doi: 10.1007/s00412-015-0558-1) both in immunofluorescence and western Blot. It is surprising that this study is not cited and that no attempt were made to assess H3K9me3 levels—e.g., via immunofluorescence—in this work to try to prove their hypothesis. Maybe authors tried, but there is no mention in the current text.
- I am aware that authors included the sentence “To validate this pathway, future research should investigate the mechanistic link between HP1 and histone modifications”.
However, I strongly encourage the authors to either perform H3K9me3 immunostaining—especially in tissues showing male traits upon siHP1-1a/b treatment, such as first antennae—or discuss this limitation more explicitly and cite the relevant literature. If experimental validation is not possible, I suggest that at least this is discussed in a more complete way with the correspondent citations, or alternatively, that speculative sentences about H3K9m3 should then be removed from the discussion.
In conclusion, the manuscript presents valuable findings with high potential, but I believe these suggestions should be addressed before proceeding with publication.
Author Response
Reviewer 2
I appreciate that the editor has consider my participation as reviewer in this manuscript.
This manuscript presents a compelling study suggesting that environmental cues trigger male production in Daphnia magna by lifting HP1-1-mediated silencing of the male-determining gene Dsx1. Knockdown of HP1-1 in females leads to the emergence of male traits, revealing its role in repressing male development under normal conditions.
Overall, this is a very interesting work, and is generally well designed and executed, however, I have some comments. Since it is difficult to give specific details as the MS is sent already formatted without line numbers, I will give the comments for each block.
Response: We sincerely thank Reviewer 2 for the positive and encouraging feedback. We are glad that the reviewer found our study compelling and well executed. We also appreciate the constructive comments provided, which helped us improve the clarity and impact of the manuscript. Below, we address each point in detail.
Introduction:
The introduction is too brief, particularly regarding aspects of Daphnia biology and sex determination, which may not be familiar to all readers.
Response: We thank the reviewer for pointing out that the original Introduction lacked sufficient background on Daphnia biology and sex determination. In response, we have revised the Introduction to provide more accessible and detailed explanations, particularly for readers who may be unfamiliar with the Daphnia model system. These revisions include clearer descriptions of environmental sex determination (ESD) in Daphnia magna, the role of the Dsx1 gene, and how male development is regulated under stress conditions.
- Please expand on the sentence beginning “DM-domain transcription factor was first identified...” (line 8 of the introduction). I assume the authors are referring to the DM transcription factor family. Additional background on this family and its role in sex determination across species would help readers better contextualize the study.
Response (lines 40–44): We thank the reviewer for the suggestion to provide more context about the DM-domain transcription factor family. In response, we have revised the Introduction to briefly explain the evolutionary conservation and functional significance of DM-domain proteins in sex determination across diverse animal taxa. This addition will help readers unfamiliar with the field better understand the relevance of Dsx1 as a male-determining gene in Daphnia magna.
- A clearer explanation of sex biology in Daphnia species is needed. Specifically, please clarify that males are genetically identical clones of their mothers (page 2, line 5). While this may seem obvious to Daphnia researchers, it is not for a broader audience.
Response (lines 49–55): Thank you for this helpful suggestion. We have revised the relevant section in the Introduction to clearly state that males in Daphnia magna are genetically identical to their mothers and sisters, as they are produced clonally from unfertilized eggs. This clarification should improve accessibility for readers unfamiliar with Daphnia biology.
- The sentence, “In contrast, Dsx1 is silenced in females throughout their lifespan, suggesting that environmental cues unlock the silencing of this male-determining gene,” needs a citation. Otherwise, this claim should be removed or rephrased. Also, if there is any work that has measure the expression levels of Dsw1 in juvenils versus adults, it should also be cited.
Response (lines 63–66): We thank the reviewer for this important suggestion. The statement regarding lifelong silencing of Dsx1 in females is now supported by the following citation:
“Nong, Q.D.; Mohamad Ishak, N.S.; Matsuura, T.; Kato, Y.; Watanabe, H. Mapping the Expression of the Sex Determining Factor Doublesex1 in Daphnia Magna Using a Knock-in Reporter. Sci Rep 2017, 7, 13521, doi:10.1038/s41598-017-13730-4.”
This study used a knock-in reporter system to directly visualize Dsx1 expression across developmental stages in both sexes, and clearly showed its persistent absence in females, including juveniles and adults.
- The sentence “It plays a crucial role in epigenetic gene silencing during development [9–11].” needs more explanation and detail about the roles of HP1. Please elaborate on HP1’s known roles and specify the species used in the cited studies.
Response (lines 69–73): Thank you for your helpful suggestion. We have revised the sentence to more clearly explain the role of HP1 in transcriptional repression. We now describe how HP1 mediates gene silencing by recognizing histone H3 lysine 9 methylation (H3K9me) and promoting heterochromatin formation. We have also specified that this function has been demonstrated across multiple species, including Drosophila melanogaster, mammals, and Arabidopsis thaliana, as supported by references [13–16] in the revised manuscript.
Materials and methods
The manuscript does not explain how the authors determined which eggs were “destined to develop as females” or “as males.” This phrasing appears several times and needs clarification.
From the current text, it appears that experiments were conducted blindly, and phenotypic sex was determined afterward. If this is the case, please clarify the environmental or experimental conditions (e.g., photoperiod, crowding, nutrient deprivation) under which males were induced.
If a specific male-inducing protocol was used—such as exposure to Methyl Farnesoate (MF)—this must be stated explicitly. Please include references (e.g., Olmstead & LeBlanc, 2002) and describe the details of the protocol, including whether it achieved 100% male offspring, to support reproducibility.
Response (lines 93–104): We thank the reviewer for this important comment. In our study, male offspring were reliably induced by exposing gravid females to the Juvenile Hormone agonist Fenoxycarb (a functional analog of Methyl Farnesoate) approximately 10 hours before the known sex-determination window (52–56 hours post-oviposition), following the methodology established by Olmstead and LeBlanc (2002) and previous our study. This timing was chosen based on prior knowledge of the temporal sensitivity of sex determination in adult Daphnia. To confirm the reliability of male induction, we performed phenotypic checks on the earliest and latest embryos from each treatment batch. Only when all tested offspring developed as males were the embryos from that batch used for downstream experiments. This verification step ensured that we used exclusively male-destined embryos for analyses requiring known sex identity at the embryonic stage. We have added this methodological clarification and appropriate citation to the revised Materials and Methods section for clarity and reproducibility.
Results
Figure 1 and Table 1: It is unclear whether the siHP1-1a individuals survived the treatment. Table 1 notes they did not swim, but does this mean they died immediately, or were they alive but immobile? Were they able to reach a juvenile stage? Any additional observations—especially regarding gonadal development—would help clarify this point.
Response (Table, 1, lines 256–259): Thank you for your thoughtful comment. We have revised Table 1 to better reflect the post-treatment condition of siHP1-1a and siHP1-1b individuals. Upon reanalysis, we confirmed that these embryos hatched normally and developed into juvenile-stage individuals, but they were immobile and did not exhibit swimming behavior. All of them died shortly after reaching the juvenile stage. Thus, we have now classified them as “immobile juveniles” in the updated table. Importantly, due to their early death, we were unable to dissect or assess their gonadal structures. However, these immobile juveniles consistently exhibited strong Dsx1-mCherry expression in the first antennae, suggesting masculinization at the tissue level. We have included this clarification in both the revised Results section and Table 1.
Figure 2: Supplementary Figure 2 contains important data and should be moved into the main figure set to improve accessibility and understanding.
Response (Supplementary Figure 5): We appreciate the reviewer’s suggestion to highlight the data presented in Supplementary Figure 2. Upon further consideration, we agree that this dataset—examining the effects of HP1-1 knockdown in males—is informative, particularly for assessing sex-specific functional roles. However, since HP1-1 knockdown in males did not lead to any significant change in Dsx1 expression (despite a substantial reduction in HP1-1 transcript levels), the physiological relevance remains uncertain. Given that our knockdown approach does not achieve complete loss-of-function and that the endogenous role of HP1-1 in males is still unclear, we have opted to retain these results in the supplementary materials to avoid overinterpreting incomplete data. To clarify their importance, we have updated the figure designation to Supplementary Figure 5 and cited it explicitly in the main text, along with a brief explanatory note. We hope this maintains clarity for readers while appropriately contextualizing the limitations of the current result.
Figures 3 and 4: Please explain the rationale for analyzing females at 48h post-injection (stage 12) and males at 30h (stage 7). If these time points differ for a biological reason, this should be clearly justified. Otherwise, a standardized time point would have been the ideal experimental procedure to improve comparability.
Response (lines 156–161, lines 182–186): Thank you for raising this important point. We performed time-course observations for both male and female embryos and selected these time points based on key biological markers and technical feasibility. For females injected with siRNA, we selected 48 hours post-injection (stage 12) because this is when the sexually dimorphic elongation of the first antenna becomes clearly detectable if Dsx1 is de-repressed. Moreover, Dsx1-mCherry expression is robust and spatially localized at this stage, enabling reliable assessment of both morphological and molecular phenotypes. For males injected with HP1-1 expression plasmid, we selected 30 hours post-injection (stage 7). This timing corresponds to the early developmental stage when sex-specific differences in dsx1 expression begin to be observed clearly. Given that HP1-1 overexpression was initiated at the one-cell stage and is transient, the 30 h time point was chosen to capture its regulatory effect on dsx1 before HP1-1 expression declines in later stages and its impact is no longer detectable. We have clarified this rationale in the revised Materials and Methods section.
Figure 3 Legend: There is a typographical error. It currently reads: “The scale bar れ…” Please correct this.
Response: We thank the reviewer for pointing out this typographical error. It has now been corrected in the revised manuscript.
Discussion
Please, homogenize the format. Introduction uses the terms GSD and ESD, however, they are not used in the discussion.
Response (lines 315–322, lines 379–391): We appreciate the reviewer’s suggestion to improve consistency across the manuscript. In response, we revised the opening section of the Discussion to more explicitly incorporate the terms “GSD” (genetic sex determination) and “ESD” (environmental sex determination), aligning it with the terminology introduced earlier in the manuscript. Specifically, we now contrast the initiating cues and downstream mechanisms between GSD and ESD systems, and we highlight how our findings on HP1-1-mediated repression in Daphnia magna fit into this broader conceptual framework.
In addition, we have revised the Discussion section to compare the HP1-1–mediated silencing of dsx1 in Daphnia magna with mechanisms used in GSD systems. While GSD often involves activation of sex-determining genes, repression of the opposite pathway also plays an essential role. By emphasizing this shared reliance on both activation and repression, we contextualize our findings within a broader regulatory logic common to both GSD and ESD systems. These revisions help ensure terminological consistency and improve the clarity and general relevance of the discussion.
Although the conclusions of the work are very interesting, some of the discussion seems speculative.
For example, authors claim:
- “One possible explanation for this discrepancy is that transcriptional activation of Dsx1 in male-destined embryos may inhibit H3K9 trimethylation, preventing HP1-1 from binding to the Dsx1 locus.”,
However, this not directly supported by experimental data in the present MS. There is no evidence presented that H3K9me3 levels were examined in the various experimental groups (both in females and males).
This is particularly relevant given that this histone modification (H3K9m3) has previously been observed in Daphnia magna, specifically in male gonadas (doi: 10.1007/s00412-015-0558-1) both in immunofluorescence and western Blot. It is surprising that this study is not cited and that no attempt were made to assess H3K9me3 levels—e.g., via immunofluorescence—in this work to try to prove their hypothesis. Maybe authors tried, but there is no mention in the current text.
- I am aware that authors included the sentence “To validate this pathway, future research should investigate the mechanistic link between HP1 and histone modifications”.
However, I strongly encourage the authors to either perform H3K9me3 immunostaining—especially in tissues showing male traits upon siHP1-1a/b treatment, such as first antennae—or discuss this limitation more explicitly and cite the relevant literature. If experimental validation is not possible, I suggest that at least this is discussed in a more complete way with the correspondent citations, or alternatively, that speculative sentences about H3K9m3 should then be removed from the discussion.
Response (lines 392–396): We thank the reviewer for this critical observation. We agree that our current dataset does not provide direct evidence for the involvement of H3K9me3 in Dsx1 regulation. In response to this concern, we have removed speculative statements regarding H3K9 trimethylation from the revised Discussion. Specifically, we eliminated the subsection entitled “HP1-dependent Dsx1 gene silencing in females,” which had previously discussed a potential mechanism involving transcriptional interference with H3K9me3 deposition.
Instead, we now explicitly acknowledge this limitation and emphasize the need for future studies to assess histone modification states at the Dsx1 locus to clarify the epigenetic mechanism of HP1-1–mediated silencing. We have also incorporated and cited the important prior study (2015, doi: 10.1007/s00412-015-0558-1), which demonstrated the presence of H3K9me3 in male gonads of D. magna using both immunofluorescence and western blotting. While our current study focused on embryonic stages, we agree that evaluating H3K9me3 distribution at the Dsx1 locus—particularly in tissues displaying male traits following siHP1-1a/b treatment—represents a crucial direction for future work.
In conclusion, the manuscript presents valuable findings with high potential, but I believe these suggestions should be addressed before proceeding with publication.
Response: We thank the reviewer for their overall positive evaluation and thoughtful suggestions, which have significantly improved the clarity, rigor, and interpretability of our manuscript. In the revised version, we have carefully addressed all the concerns raised, including the removal of speculative discussion regarding H3K9me3, the acknowledgment of limitations in our dataset, and the inclusion of relevant prior literature (2015, doi: 10.1007/s00412-015-0558-1) to appropriately contextualize our findings. We believe that these revisions have substantially strengthened the manuscript, and we hope it will now meet the standards for publication.

Reviewer 3 Report
Comments and Suggestions for Authors
The authors investigate the role of a chromodomain protein variant, Heterochromatin Protein 1-1 in the crustacean Daphnia magna. Their work shows that in this species, which reproduces by parthenogenesis, HP1-1 is involved in the suppression of the sex-determination gene doublesex1 in females, as knockdown of HP1-1 in female embryos leads to a strong increase in dsx1 expression and the formation of male traits. On the other hand, over expression of HP1-1 in male embryos does not lead to a dsx1 suppression and female development. Therefore, it is not clear yet what the mechanism of HP1-1 suppression of dsx1 is, and how this suppression is abolished through environmental cues. This will be an important subject for future studies.
Dear authors,
this manuscript is well written and the results are mostly clearly presented. therefore I have only some points that need to be addressed before publication. You find all of may comments in the uploaded PDF. The major points that need to be addressed are:
1) I would ask the author to please carefully revise the methods section to make sure all relevant information is provided that is needed to replicate the experiments exactly as you did it. there are important details missing, of which I have marked some.
2) you report the quantification of fluorescence intensity in the methods section, but don't report the data in the manuscript. I believe that this data would strengthen the message of your study as it would put the conclusions you make based on single images shown to the reader on a much stronger foundation based on RFI calculated across replicates and thereby also much more trustworthy. Therefore please include this valuable data either in the main manuscript or the supplement.
3)I'm struggling with the points and hypothesis you raise in the discussion. I honestly find it hard to follow and to judge how the different hypothesis you raise there might be connected or fit together in the ESD pathway and fit with your findings on HP1-1 in D. magna. For example you speculate - based of your results - that dsx1 expression is suppressed by HP1-1, and that dsx1 expression in future males prevents repressive chromatin marks which then prevent HP1 binding and dsx1 repression. but how is dsx1 then switched on in the first place? the 2nd hypothesis in this paragraph has the same issue: who was first? the hen or the egg? therefore I would ask you to carefully revise the discussion and reconsider your arguments and if they really fit the data and can be logically connected. Moreover, the introductions to the different discussion subsections are somewhat repetitive, therefore please streamline it and clean it up. moreover, a schematic that summarises the different hypothesis and factors you suggest and how they might be interconnected with Hp1 and dsx1 activation would tremendously help to understand your ideas.

Round 2
Reviewer 2 Report
Comments and Suggestions for Authors
Authors have answered to most my concerns by implementing most of the suggestions and have included essential improvements in the text and figures. Specifically, I appreciate the value of including new Figure 6. I think that the new version is significantly improved and ready for publication.